# Collegial Ensembles

**Etai Littwin   Ben Myara   Sima Sabah   Joshua Susskind   Shuangfei Zhai  Oren Golan**
Apple Inc.
{elittwin, bmyara, sima, jsusskind, szhai, ogolan}@apple.com

## Abstract

Modern neural network performance typically improves as model size increases. A recent line of research on the Neural Tangent Kernel (NTK) of over-parameterized networks indicates that the improvement with size increase is a product of a better conditioned loss landscape. In this work, we investigate a form of over-parameterization achieved through ensembling, where we define collegial ensembles (CE) as the aggregation of multiple independent models with identical architectures, trained as a single model. We show that the optimization dynamics of CE simplify dramatically when the number of models in the ensemble is large, resembling the dynamics of wide models, yet scale much more favorably. We use recent theoretical results on the finite width corrections of the NTK to perform efficient architecture search in a space of finite width CE that aims to either minimize capacity, or maximize trainability under a set of constraints. The resulting ensembles can be efficiently implemented in practical architectures using group convolutions and block diagonal layers. Finally, we show how our framework can be used to analytically derive optimal group convolution modules originally found using expensive grid searches, without having to train a single model.

## 1   Introduction

Neural networks exhibit generalization behavior in the over-parameterized regime, a phenomenon that has been well observed in practice [23, 2, 18, 17]. Recent theoretical advancements have been made to try and understand the trainability and generalization properties of over-parameterized neural networks, by observing their nearly convex behaviour at large width [13, 15]. For a wide neural network $\mathcal{F}(x)$ with parameters $\theta$ and a convex loss $\mathcal{L}$, the parameter updates $-\mu \nabla_\theta \mathcal{L}$ can be represented in the space of functions as kernel gradient decent (GD) updates $-\mu \nabla_{\mathcal{F}} \mathcal{L}$, with the Neural Tangent Kernel [10] (NTK) function $\mathcal{K}(x, x_j) = \nabla_\theta \mathcal{F}(x) \nabla_\theta^\top \mathcal{F}(x_j)$ operating on $x, x_j$:

$$\Delta \theta = -\mu \nabla_\theta \mathcal{L} \longrightarrow \Delta \mathcal{F}(x) \sim -\mu \sum_j \mathcal{K}(x, x_j) \nabla_{\mathcal{F}(x_j)} \mathcal{L} \qquad (1)$$

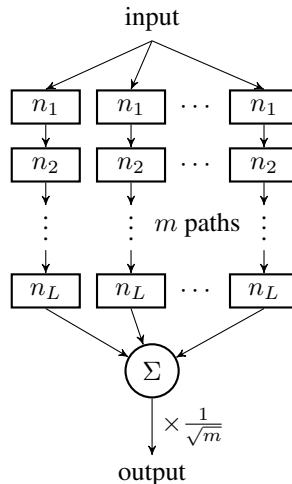

Figure 1: Collegial Ensemble

In the infinite width limit, the NTK remains constant during training, and GD reduces to kernel GD, rendering the optimization a convex problem. Hence, over parameterized models in the "large width" sense both generalize, and are simpler to optimize. In this work, we consider a different type of over-parameterization achieved through ensembling. We denote by collegial ensembles (CE) models where the output, either intermediate or final, is constructed from the aggregation of multiple identical pathways (see illustration in Fig. 1). We show that the training dynamics of CE simplify when the ensemble multiplicity is large, in a similar sense as wide models, yet at a much cheaper cost in terms of parameter count. Our results

indicate the existence of a "sweet spot" for the correct ratio between width and multiplicity, where "close to convex" behaviour does not come at the expense of size. To find said "sweet-spot", we rely on recent findings on finite corrections of gradients [7, 16], though we use them in a more general manner than their original presentation. Specifically, we use the following assumption stated informally:

**Assumption 1.** *(Informal) Denote by $\mathcal{K}$ the NTK at initialization of a fully connected ANN with hidden layer widths $n_1, ..., n_L$ and depth L. There exists positive constants $\alpha, C$ such that:*

$$Var(\mathcal{K}) \sim C(e^{\alpha \sum_{l=1}^{L} n_l^{-1}} - 1) \tag{2}$$

*where the variance is measured on the individual entries of $\mathcal{K}$, with respect to the random sampling of the weights.*

In [7] and [16], assumption 1 is proven for the on-diagonal entries of $\mathcal{K}$, in fully connected architectures. In this work, we assume it holds in a more general sense for practical architectures, with different constants of $\alpha, C$ (we defer the reader to Sec. 4 in the appendix for further empirical validation of this assumption). Since $Var(\mathcal{K})$ diminishes with width, we hypothesize that a small width neutral network behaves closer to its large width counterpart as $Var(\mathcal{K})$ decreases. Notably, similar observations using activations and gradient variance as predictors of successful initializations were presented in [6, 8]. Motivated by this hypothesis, we formulate a primal-dual constrained optimization problem that aims to find an optimal CE with respect to the following objectives:

1. **Primal (optimally smooth):** minimize $Var(\mathcal{K})$ for a fixed number of parameters.
2. **Dual (optimally compact):** minimize number of parameters for a fixed $Var(\mathcal{K})$.

The primal formulation seeks to find a CE which mimics the simplified dynamics of a wide model using a fixed budget of parameters, while Fthe dual formulation seeks to minimize the number of parameters without suffering the optimization and performance consequences typically found in the "narrow regime". Using both formulations, we find excellent agreement between our theoretical predictions and empirical results, on both small and large scale models.

Our main contributions are the following:

1. We adapt the popular over-parameterization analysis to collegial ensembles (CE), in which the output units of multiple architecturally identical models are aggregated, scaled, and trained as a single model. For ensembles with $m$ models each of width $n$, we show that under gradient flow and the L2 loss, the NTK remains close to its initial value up to a $\mathcal{O}((mn)^{-1})$ correction.

2. We formulate two optimization problems that seek to find optimal ensembles given a baseline architecture, in the primal and dual form, respectively. The optimally smooth ensemble achieves higher accuracy than the baseline, using the same budget of parameters. The optimally compact ensemble achieves a similar performance as the baseline, with significantly fewer trainable parameters.

3. We show how optimal grouping in ResNeXt [20] architectures can be derived and improved upon using our framework, without the need for an expensive search over hyper-parameters.

The remaining paper is structured as follows. In Sec. 2 we formally define collegial ensembles, and present our results for their training dynamics in the large $m, n$ regime. In Sec. 3 we present our framework for architecture search in the space of collegial ensembles, and in Sec. 4 we present further experimental results on the CIFAR-10/CIFAR-100 [11] and ImageNet [4] datasets using large scale models.

## 2 Collegial Ensembles

We dedicate this section to formally define collegial ensembles, and analyze their properties from the NTK perspective. Specifically, we would like to formalize the notion of the "large ensemble" regime, where its dynamic behaviour is reminiscent of wide single models. In the following analysis we consider simple feed forward fully connected neural network $\mathcal{F}_\mathbf{n}(x, \theta) : \mathbb{R}^{n_0} \to \mathbb{R}$, where the width

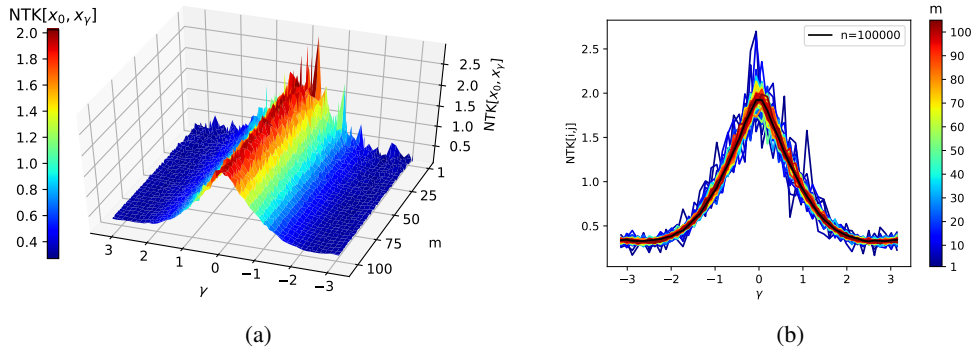

Figure 2: **Convergence of the ensemble NTK to the NMK** when increasing the number of models in the ensemble. $NTK[x_0, x_\gamma]$ is computed for both the diagonal and off-diagonal elements $x_\gamma = [\cos(\gamma), \sin(\gamma)]$ for $\gamma \in [-\pi, \pi]$. The smoother surface as $m$ increases in (a) demonstrates the convergence of the NTK. The black line in (b), computed with $\times 100$ wider model, shows that the convergence is indeed to the NMK, and the NTK mean does not depend on the width of the model. Each model in the ensemble is fully connected with $L = 4$ layers and $n = 1000$.

of the hidden layers are given by $\mathbf{n} = \{n_l\}_{l=1}^L \in \mathbb{Z}_+^L$, adopting the common NTK parameterization:

$$\mathcal{F}_{\mathbf{n}}(x, \theta) = \sqrt{\frac{1}{n_L}} \theta^L (...\phi(\sqrt{\frac{2}{n_1}} \theta^2 \phi(\sqrt{\frac{2}{n_0}} \theta^1 x))). \tag{3}$$

where $x \in \mathbb{R}^{n_o}$ is the input, $\phi(\cdot)$ is the ReLU activation function, $\theta^l$ is the weight matrix associated with layer $l$, and $\theta$ denotes the concatenation of all the weights of all layers, which are initialized iid from a standard normal distribution. Given a dataset $X = \{x_i\}_{i=1}^N$, the empirical NTK denoted by $\mathcal{K}_{\mathbf{n}}(\theta) \in \mathbb{R}^{N \times N}$ is given by $\mathcal{K}_{\mathbf{n}}(\theta) = \nabla_\theta \mathcal{F}_{\mathbf{n}}(\theta) \nabla_\theta^\top \mathcal{F}_{\mathbf{n}}(\theta)$, where $\mathcal{F}_{\mathbf{n}}(\theta) = [\mathcal{F}_{\mathbf{n}}(x_1, \theta), ..., \mathcal{F}_{\mathbf{n}}(x_N, \theta)]^\top$. Given the network $\mathcal{F}_{\mathbf{n}}$, we parameterize a space of ensemble members $\mathcal{F}^e(\Theta)$ by a multiplicity parameter $1 \leq m$ and a width vector $\mathbf{n}$, such that:

$$\mathcal{F}^e(\Theta) = \frac{1}{\sqrt{m}} \sum_{j=1}^m \mathcal{F}_{\mathbf{n}}(\theta_j), \quad \mathcal{K}^e(\Theta) = \frac{1}{m} \sum_{j=1}^m \mathcal{K}_{\mathbf{n}}(\theta_j) \tag{4}$$

where $\Theta = [\theta_1^\top ... \theta_m^\top]^\top$ is the concatenation of the weights of all the ensembles, and $\mathcal{K}^e(\Theta) = \nabla_\Theta \mathcal{F}^e \nabla_\Theta^\top \mathcal{F}^e$ is the NTK of the ensemble. Plainly speaking, the network $\mathcal{F}_{\mathbf{n}}$ defines a space of ensembles given by the scaled sum of $m$ neural networks of the same $\mathcal{F}_{\mathbf{n}}$ architecture, with weights initialized independently from a normal distribution. Since each model $\mathcal{F}_{\mathbf{n}}(\theta_j)$ in the ensemble is architecturally equivalent to $\mathcal{K}_{\mathbf{n}}(\theta)$, it is easy to show that the infinite width kernel is equal for both models: $\mathcal{K}_\infty = \lim_{min(\mathbf{n}) \to \infty} \mathcal{K}_{\mathbf{n}}(\theta) = \lim_{min(\mathbf{n}) \to \infty} \mathcal{K}^e(\Theta)$. We define the Neural Mean Kernel (NMK) $\mathcal{K}_{\mathbf{n}}^\infty$ as the mean of the empirical NTK:

$$\mathcal{K}_{\mathbf{n}}^\infty = \mathbb{E}_\theta[\mathcal{K}_{\mathbf{n}}(\theta)]]. \tag{5}$$

The NMK is defined by an expectation over the normally distributed weights, and does not immediately equal the infinite width limit of the NTK given by $\mathcal{K}_\infty$. The following Lemma stems from the application of the strong law of large numbers (LLN):

**Lemma 1** (Infinite ensemble). *The following holds:*

$$\mathcal{K}^e(\Theta) \xrightarrow{a.s} \mathcal{K}_{\mathbf{n}}^\infty \quad \text{as } m \to \infty. \tag{6}$$

We defer the reader to Sec. 5 in the appendix for the full proof. While both $\mathcal{K}_{\mathbf{n}}^\infty$ and $\mathcal{K}_\infty$ do not depend on the weights, they are defined differently. $\mathcal{K}_{\mathbf{n}}^\infty$ is defined by an expectation over the weights, and depends on the width of the architecture, whereas $\mathcal{K}_\infty$ is defined by an infinite width limit. However, empirical observation using Monte Carlo sampling, as presented in Fig. 2, show little to no dependence of the NMK on the widths $\mathbf{n}$. Moreover, we empirically observe that $\mathcal{K}_{\mathbf{n}}^\infty \sim \mathcal{K}_\infty$ (Note that similar observations have been reported in [10]). We next show that under gradient flow, $\mathcal{K}^e(\Theta)$ remains close to its initial value for the duration of the optimization process when $mn$ is large. Given

the labels vector $\mathbf{y} \in \mathbb{R}^N$ and the $L_2$ cost function at time $t$, $\mathcal{L}_t = \frac{1}{2}\|\mathcal{F}^e(\Theta_t) - \mathbf{y}\|_2^2$, under gradient flow with learning rate $\mu$, the weights evolve over continuous time according to $\dot{\Theta}_t = -\mu \nabla_{\Theta_t}^\top \mathcal{L}_t$. The following theorem gives an asymptotic bound on the leading correction of the ensemble NTK over time. For simplicity, we state our result for constant width networks.

**Theorem 1** (NTK evolution over time). *Assuming analytic activation functions $\phi(\cdot)$ with bounded derivatives of all orders, and the $L_2$ cost function , it holds for any finite $t$:*

$$\mathcal{K}^e(\Theta_t) - \mathcal{K}^e(\Theta_0) \sim \mathcal{O}_p(\frac{1}{mn}) \tag{7}$$

*where the notation $x_n = \mathcal{O}_p(y_n)$ states that $x_n/y_n$ is stochastically bounded.*

We defer the reader to Sec. 5 in the appendix for the full proof, as well as empirical validation of The. 1. Large collegial ensembles therefore refer to a regime where $mn$ is large. In the case of infinite multiplicity, optimization dynamics reduces to kernel gradient descent with $\mathcal{K}_\mathbf{n}^\infty$, rather than $\mathcal{K}_\infty$ as the relevant kernel function. A striking implication arises from Theorem 1. The total parameter count in the ensemble is linear in $m$, and quadratic in $n$, hence it is much more parameter efficient to increase $m$ rather than $n$. Since the "large" regime depends on both $n$ and $m$, CE possess an inherent degree of flexibility in their practical design. As we show in the following section, this increased flexibility allows the design of both parameter efficient ensembles, and better performing ensembles, when compared with the baseline model.

## 3 Efficient Ensembles

In this section, we use Proposition 1 to derive optimally smooth and compact ensembles. We parameterize the space of ensembles using $m, \mathbf{n}$, and a baseline architecture $\mathcal{F}_{\tilde{\mathbf{n}}}$, where $\tilde{\mathbf{n}}$ is the width vector of the baseline model. Denote by $\beta(\mathbf{n})$ the total parameter count in $\mathcal{F}_\mathbf{n}$, we define parameter efficiency $\rho(m, \mathbf{n})$ by the ratio between the parameter count in the baseline model $\beta_s \triangleq \beta(\tilde{\mathbf{n}})$, and the parameter count in the ensemble given by $\beta_e \triangleq m\beta(\mathbf{n})$:

$$\rho(m, \mathbf{n}) \triangleq \frac{\beta_s}{\beta_e} = \frac{\beta_s}{m\beta(\mathbf{n})}. \tag{8}$$

Using Proposition. 1, the variance of $\mathcal{K}_\mathbf{n}$ as a function of widths $\mathbf{n}$ and depth $L$, is given by:

$$Var\big(\mathcal{K}_\mathbf{n}(\theta)\big) \sim (e^{\alpha \sum_{l=1}^L n_l^{-1}} - 1). \tag{9}$$

for some value of $\alpha$.

**Primal formulation:** We cast the primal objective as an optimization problem, where we would like to find parameters $m^p, \mathbf{n}^p$ that correspond to the smoothest ensemble:

$$m^p, \mathbf{n}^p = \underset{m, \mathbf{n}}{\arg \min} Var\big(\mathcal{K}^e(\Theta)\big) \quad s.t \quad \rho(m, \mathbf{n}) = 1. \tag{10}$$

Since the weights for each model are sampled independently, it holds that:

$$Var\big(\mathcal{K}^e(\Theta)\big) = \sum_{j=1}^m \frac{1}{m^2} Var\big(\mathcal{K}_\mathbf{n}(\theta_j)\big) = \frac{(e^{\alpha \sum_{l=1}^L n_l^{-1}} - 1)}{m}. \tag{11}$$

Equating the parameter count in both models to maintain a fixed efficiency, we can derive for each $\mathbf{n}$ the number of the models $m^p(\mathbf{n})$ in the primal formulation:

$$m^p(\mathbf{n}) = \frac{\beta_s}{\beta(\mathbf{n})} \quad \longrightarrow \quad \mathbf{n}^p = \underset{\mathbf{n}}{\arg \min} \left[ \frac{(e^{\alpha \sum_{l=1}^L n_l^{-1}} - 1)}{m^p(\mathbf{n})} \right]. \tag{12}$$

The optimal parameters $\mathbf{n}^p$ can be found using a grid search.

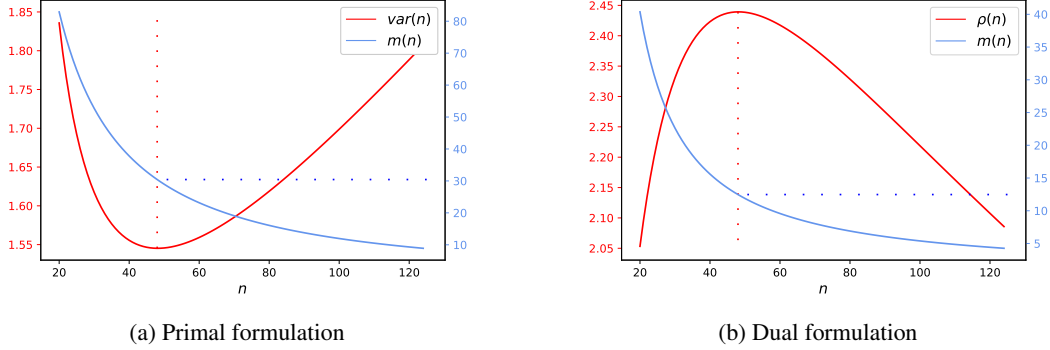

(a) Primal formulation          (b) Dual formulation

Figure 3: **Primal and dual objective curves** for a baseline feedforward fully connected network with $L = 6$ layers, $\tilde{n} = 500$, and $n_0 = 748$. (a) The minimizer of the primal objective (red) is achieved for $n = 48$ and $m(48) \approx 30$. (b) The maximizer of the dual objective (red) is achieved for $n = 48$ and $m(48) \approx 12$, achieving an efficiency value $\rho(48) \approx 2.45$.

**Dual formulation:** The dual formulation can be cast as an optimization problem, with the following objective:

$$m^d, \mathbf{n}^d = \underset{m,\mathbf{n}}{\arg\max}\, \rho(m, \mathbf{n}) \quad s.t \quad Var\big(\mathcal{K}^e(\Theta)\big) = Var\big(\mathcal{K}_{\tilde{\mathbf{n}}}(\theta)\big). \quad (13)$$

Matching the smoothness criterion using Eq. 11, we can derive for each $\mathbf{n}$ the number of models $m^d(\mathbf{n})$ in the dual formulation:

$$m^d(\mathbf{n}) = \frac{\big(e^{\alpha \sum_{l=1}^{L} n_l^{-1}} - 1\big)}{\big(e^{\alpha \sum_{l=1}^{L} \tilde{n}_l^{-1}} - 1\big)} \quad \longrightarrow \quad \mathbf{n}^d = \underset{\mathbf{n}}{\arg\max} \left[\frac{1}{m^d(\mathbf{n})} \frac{\beta_s}{\beta(\mathbf{n})}\right]. \quad (14)$$

Ideally, we can find $m^d, \mathbf{n}^d$ such that the total parameter count in the ensemble is considerably reduced. Equating the solutions for both the primal and dual problems in Eq. 12 and Eq. 14, it is straightforward to verify that $\mathbf{n}^d = \mathbf{n}^p$, implying strong duality. Therefore, the primal and dual solutions differ only in the optimal multiplicity $m(\mathbf{n})$ of the ensemble. Both objectives are plotted in Fig. 3 using a feedforward fully connected network baseline with $L = 6$ and constant width $\tilde{n} = 500$.

Note that the efficient ensembles framework outlined in this section can readily be applied with different efficiency metrics. For instance, instead of using the parameter efficiency, one could consider the FLOPs efficiency (see Appendix Sec. 3).

## 4 Experiments

In the following section we conduct experiments to both validate our assumptions, and evaluate our framework for efficient ensemble search. Starting with a toy example, we evaluate the effect of $Var(\mathcal{K})$ and $\beta_e$ on test performance using fully connected models trained on the MNIST dataset. For the latter experiments, we move to larger scale models trained on CIFAR-10/100 and the ImageNet [4] datasets.

### 4.1 Ablation Study – MNIST

An effective way to improve the performance of a neural network is to increase its size. Recent slim architectures, such as ResNeXt, demonstrate it possible to maintain accuracy while significantly reducing parameter count. In Fig. 4 we provide further empirical evidence that capacity of a network by itself is not a good predictor of performance, when decoupled from other factors.

Specifically, we demonstrate strong correlation between the empirical test error and the variance $Var(\mathcal{K})$, while $\beta_e$ is kept constant (primal). On the other hand, increasing $\beta_e$ while keeping $Var(\mathcal{K})$ constant (dual) does not improve the performance. For both experiments we use as a baseline a fully connected model with $L = 6$ layers and width $\tilde{n} = 200$ for each layer. The width of a layer for each of the $m$ models in the ensemble is $n$. Each ensemble was trained on MNIST for 70 epochs with the Adam optimizer, and the accuracy was averaged over 100 repetitions.

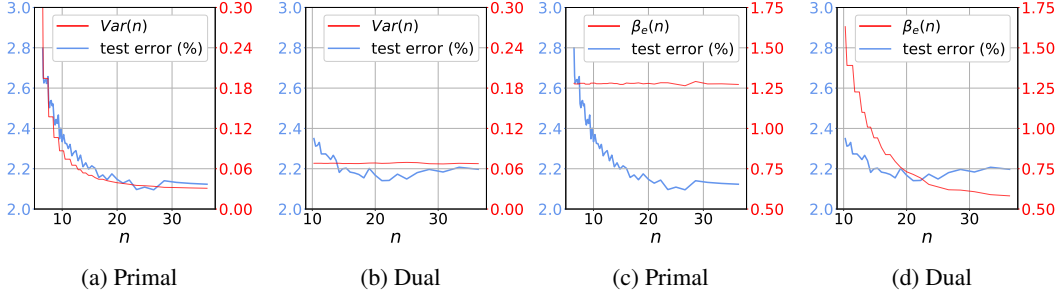

|     (a) Primal     |     (b) Dual     |     (c) Primal     |     (d) Dual     |

Figure 4: **Decoupling capacity and variance**. The error (blue) is highly correlated with $Var(\mathcal{K})$, and less sensitive to $\beta_e$. (a) and (b) show the theoretical variance of the model correlates well with accuracy. (c) and (d) show the corresponding number of parameters $\beta_e$. Decreasing the variance (a) improves performance when $\beta_e$ is fixed (c). Increasing $\beta_e$ significantly (d) without reducing the variance (b) can cause degradation in performance due to overfitting.

## 4.2 Aggregated Residual Transformations

ResNeXt [20] introduces the concept of aggregated residual transformations utilizing group convolutions, which achieves better parameter/capacity trade off than its ResNet counterpart. In this paper, we view ResNeXt as a special case of CE, where the ensembling happens at the block level. We hypothesize that the optimal blocks identified with CE will lead to an overall better model when stacked up, and by doing so we get the benefit of factorizing the design space of a network to modular levels. See Algorithm 2 for the detailed recipe.

For these experiments, we use both the **CIFAR-10/100** and the **ImageNet** datasets following the implementation details described in [20]. We also report results on **ImageNet64×64** and **ImageNet32×32**, datasets introduced in [3] that preserve the number of samples and classes of the original ImageNet-1K [4], but downsample the image resolutions to $64\times64$ and $32\times32$ respectively (see Appendix Sec. 1).

**Fitting $\alpha$ to a ResNet block.** The first step in the optimization required for both the primal and dual objectives, is to approximate the $\alpha$ parameter in Eq. 9. For convolutional layers, the coefficient multiplying $\alpha$ becomes $\sum_{l=1}^{L} \text{fan-in}_l^{-1}$ where fan-in$_l$ is the fan-in of layer $l$. Following Algorithm 1, we approximate the $\alpha$ corresponding to a ResNet block parametrized by $\mathbf{n} = [n, n]^\top$ as depicted in Fig. 5. We compute a Monte Carlo estimate of the second moment of one diagonal entry of the NTK matrix for increasing width $n \in [\![1, 256]\!]$ and fixed $n_{in}=n_{out}=256$. For simplicity, we fit the second moment normalized by the squared first moment, given by $e^{\alpha \sum_{l=1}^{L} \text{fan-in}_l^{-1}}$, which can easily be fitted with a first degree polynomial when considering its natural logarithm. We find $\alpha \approx 1.60$ and show the fitted second moment in Appendix Fig. 2.

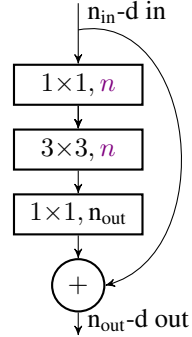

Figure 5

### 4.2.1 CIFAR-10/100

**Primal formulation.** As a baseline architecture, we use a $1\times128$d ResNet, following the notations of [20] section 5.1. Following Algorithm 2, we compute $m(n)$ for $n \in [\![1, 128]\!]$ and find the optimum $n^p = 10$ and $m^p \approx 37$, after adjusting $m^p$ to match the number of parameters of the baseline and account for rounding errors and different block topology approximations. As can be seen in Table 1a, the model achieving the primal optimum, $37\times10$d, attains better test error on CIFAR-10/100 than the ResNeXt baseline $3\times64$d at a similar parameter budget. We also report results for a wider baseline $8\times64$d from [20] and show similar trends. The test error for multiple models sitting on the primal curve is depicted in Fig. 6a for CIFAR-100 and Appendix Fig. 1 for CIFAR-10. Test errors are averaged over the last 10 epochs over 10 runs.

**Dual formulation.** Using the same ResNet base block as for the primal experiment, thus using the same fitted $\alpha$, we compute the optimal $n^d$ and $m^d$ maximizing the parameter efficiency curve $\rho$ and find the same $n$ as the primal, $n^d = 10$, and $m^d \approx 10$. The resulting ResNeXt network has 3.3 times fewer parameters than the baseline and achieves similar or slightly degraded performance on

CIFAR-10/100 as shown in Table 1b. The efficiency curve $\rho$ depicted in red in Fig. 6b is constructed using a single ResNet block topology and with non integer numbers for $m$ as described above. Thus it only approximates the *real* parameter efficiency, explaining why some models in the close vicinity of the optimum have a slightly higher real efficiency as can be seen in Table 1b. The test error for multiple models sitting on the dual curve is depicted in Fig. 6b for CIFAR-100 and Appendix Fig. 1 for CIFAR-10.

### 4.2.2 ImageNet

**Primal formulation.** Following [20], we use ResNet-50 and ResNet-101 as baselines and report results in Table 2a. Our ResNet-50 based optimal model, $12\times10$d, obtains slightly better top-1 and top-5 errors than the baseline $32\times4$d reported in [20]. This is quite remarkable given that [20] converged to this architecture via an expensive grid search. Our ResNet-101 based optimal model achieves a significantly better top-1 and top-5 error than the ResNet-101 baseline, and a slightly higher top-1 and top-5 error than the ResNeXt baseline $32\times4$d.

**Dual formulation.** Using ResNet-50 and ResNet-101 as baselines, we find models that achieve similar top-1 and top-5 errors with significantly less parameters. Detailed results can be found in Table 2b.

**Implementation details.** We follow [20] for the implementation details of ResNet-50, ResNet-101 and their ResNeXt counterparts. We use SGD with 0.9 momentum and a batch size of 256 on 8 GPUs (32 per GPU). The weight decay is 0.0001 and the initial learning rate 0.1. We train the models for 100 epochs and divide the learning rate by a factor of 10 at epoch 30, 60 and 90. We use the same data normalization and augmentations as in [20] except for lighting that we do not use.

## 5   Related Work

Various forms of multi-pathway neural architectures have surfaced over the years. In the seminal AlexNet architecture [12], group convolutions were used as a method to distribute training on multiple GPUs. More recently, group convolutions were popularized by ResNeXt [20], empirically demonstrating the benefit of aggregating multiple residual branches. In [24], a channel shuffle unit was introduced in order to promote information transfer between different groups. In [9] and [19], the connections between pre-defined set of groups are learned in a differentiable manner, and in [25], grouping is achieved through pruning of full convolutional blocks. In a seemingly unrelated front, the theoretical study of wide neural networks has seen considerable progress recently. A number of papers [13, 21, 1, 5, 14] have followed in the footsteps of the original NTK paper [10]. In [13], it is shown that wide models of any architecture evolve as linear models, and in [21], a general framework for computing the NTK of a broad family of architectures is proposed. Finite width corrections to the NTK are derived in [7, 15, 5]. In this work, we extend the "wide" regime to the multiplicity dimension, and show two distinct regimes where different kernels reign. We then use finite corrections of NTK to formulate two optimality criterions, and demonstrate their usefulness in predicting efficient and performant ensembles.

---

**Algorithm 1:** Fitting $\alpha$ per architecture

**Input:** Baseline module with $\mathbf{n}=\{n_l\}_{l=1}^{L}$, a set of width ratios $\{r_j\}_{j=1}^{R}$, $T$, samples $\{x_1, x_2\}$.

**Output:** Fitted $\alpha$.

1  **for** $j = 1, ..., R$ **do**
2       Construct module $\mathbf{n}_j = \{r_j \times n_l\}_{l=1}^{L}$.
3       **for** $t = 1, ..., T$ **do**
4           Sample weights of $\mathbf{n}_j$.
5           Compute $\mathcal{K}_{\mathbf{n_j}}(x_1, x_2)$.
6       Estimate $Var(\mathcal{K}_{\mathbf{n_j}})$.
7  Fit $\alpha$ using Eq. 9.

---

**Algorithm 2:** Find Optimal CE module

**Input:** Baseline module with $\tilde{\mathbf{n}}=\{n_l\}_{l=1}^{L}$, a set of width ratios $\{r_j\}_{j=1}^{R}$, $T$, samples $\{x_1, x_2\}$, $\beta_s$.

**Output:** optimal $\mathbf{n}^\star, m(\mathbf{n}^\star)$.

1  Fit $\alpha$ using Algorithm 1.
2  **if** *Primal* **then**
3       find $\mathbf{n} = \mathbf{n}^p$ and $m = m^p$ using Eq. 12.
4  **else if** *Dual* **then**
5       find $\mathbf{n} = \mathbf{n}^d$ and $m = m^d$ using Eq. 14.
6  Correct $\mathbf{n}$ and $m$ to nearest integer values.

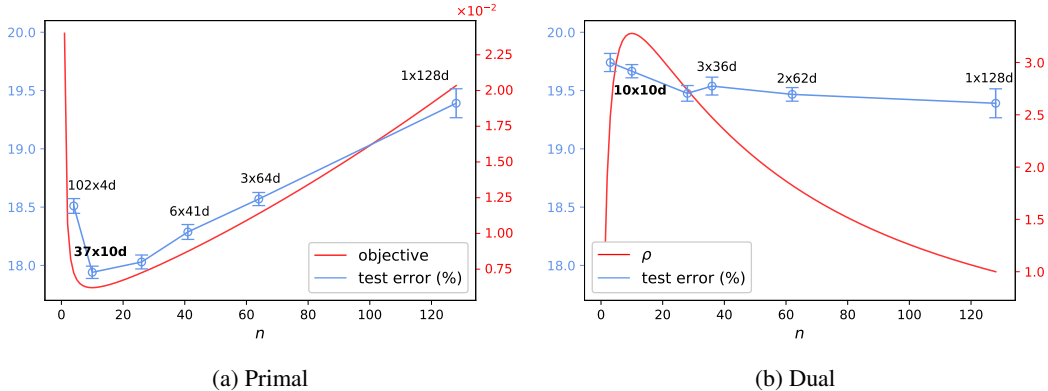

|          | (a) Primal | (b) Dual |
|----------|------------|----------|

Figure 6: **Test errors (in %) on CIFAR-100.** Clear correlation is observed between test error and the primal curve. (a) Different models sitting on the primal curve. The model achieving the primal minimum, 37×10d, achieves the best error. (b) Different models sitting on the dual curve. The model 10×10d achieves the dual maximum ($\rho \approx 3.3$) and a test error comparable to the baseline with 3.3 times fewer parameters. Results are reported over 10 runs and shown with standard error bars.

| Model | Params | C10 | C100 |
|-------|--------|-----|------|
| 1×128d [20][†] | 13.8M | 4.08 | 19.39 |
| 3×64d [20][†] | 13.3M | 3.96 | 18.57 |
| 37×10d (Ours) | 13.7M | 3.82 | **17.94** |
| 28×12d (Ours)[‡] | 12.9M | **3.74** | 18.05 |
| Wide ResNet [22] | 36.5M | 4.17 | 20.50 |
| 1×226d | 36.3M | 3.88 | 18.36 |
| 8×64d [20] | 34.4M | **3.65** | 17.77 |
| 101×10d (Ours) | 36.3M | **3.65** | **17.44** |

(a) Primal

| Model | $\rho$ | Params | C10 | C100 |
|-------|--------|--------|-----|------|
| 1×128d [20][†§] | 1 | 13.8M | 4.75 | 20.74 |
| 10×10d (Ours)[§] | **3.3** | **4.22M** | 4.70 | 20.84 |
| 1×128d [20][†] | 1 | 13.8M | 4.08 | 19.39 |
| 10×10d (Ours) | 3.3 | 4.22M | 4.21 | 19.67 |
| 8×12d (Ours)[‡] | **3.3** | **4.19M** | 4.20 | 19.58 |
| Wide ResNet [22] | 1 | 36.5M | 4.17 | 20.50 |
| 1×226d | 1 | 36.3M | 3.88 | 18.36 |
| 2×64d [20] | 4.0 | 9.13M | 4.02 | - |
| 14×10d (Ours) | **6.5** | **5.63M** | 4.01 | 19.04 |
| 6×25d (Ours) | 5.0 | 7.26M | 3.94 | 18.92 |
| 3×58d (Ours) | 3.1 | 11.6M | 3.87 | 18.75 |
| 2×98d (Ours) | 2.1 | 17.4M | 3.99 | 18.32 |

(b) Dual

Table 1: **Primal and dual results for ResNeXt-29 baselines on CIFAR-10/100**. Test errors (in %) for CIFAR-10 (C10) and CIFAR-100 (C100) along with model sizes are reported. All models are variants of ResNeXt-29 except for Wide ResNet. (a) The optimally smooth models, 37×10d and 101×10d, surpass the baselines with the same number of parameters. (b) The optimally compact models only use a fraction of the parameters, yet attain similar or slightly degraded test errors. $\rho$ indicates the parameter efficiency. [†] indicates we reproduced results on baseline architectures from the cited paper, [‡] indicates models in the close vicinity of the optima. Results are averaged over 10 runs. Models were trained on 8 GPUs unless indicated by [§], in which case they were trained on a single GPU.

# 6  Conclusion

Understanding the effects of model architecture on training and test performance is a longstanding goal in the deep learning community. In this work we analyzed collegial ensembling, a general technique used in practice where multiple and functionally identical pathways are aggregated. We showed that collegial ensembles exhibit two distinct regimes of over-parameterization, defined by large width and large multiplicity, with two distinct kernels governing the dynamics of each. In between these two regimes, we introduced a framework for deriving optimal ensembles in a sense of minimum capacity or maximum trainability. Empirical results on practical models demonstrate the predictive power of our framework, paving the way for more principled architecture search algorithms.

|  | Params | Top-1 error | Top-5 error |
|---|---|---|---|
| ResNet-50, $1\times64$d [20][†] | 25.6M | 23.93 | 7.11 |
| ResNeXt-50, $32\times4$d [20][†] | 25.0M | 22.42 | 6.36 |
| ResNeXt-50, $12\times10$d (Ours) | 25.8M | **22.37** | **6.30** |
| ResNeXt-50, $15\times8$d (Ours)[‡] | 25.1M | 22.39 | 6.38 |
| ResNet-101, $1\times64$d [20][†] | 44.5M | 22.32 | 6.25 |
| ResNeXt-101, $32\times4$d [20][†] | 44.2M | **21.01** | **5.72** |
| ResNeXt-101, $12\times10$d (Ours) | 45.5M | 21.16 | 5.74 |
| ResNeXt-101, $15\times8$d (Ours)[‡] | 44.2M | 21.20 | 5.76 |

(a) Primal

|  | $\rho$ | Params | Top-1 error | Top-5 error |
|---|---|---|---|---|
| ResNet-50, $1\times64$d [20][†] | 1 | 25.6M | 23.93 | **7.11** |
| ResNeXt-50, $3\times23$d (Ours) | 1.3 | 19.4M | **23.80** | **7.11** |
| ResNeXt-50, $4\times16$d (Ours) | **1.5** | **17.1M** | 24.00 | **7.11** |
| ResNet-101, $1\times64$d [20][†] | 1 | 44.5M | 22.32 | 6.25 |
| ResNeXt-101, $3\times23$d (Ours) | 1.4 | 32.9M | **22.06** | **6.11** |
| ResNeXt-101, $5\times12$d (Ours) | **1.7** | **25.8M** | 22.30 | 6.27 |

(b) Dual

Table 2: **Primal and dual results for ResNet baselines on ImageNet.** Top-1 and top-5 errors (in %) and model sizes are reported. $\rho$ indicates the parameter efficiency, [†] indicates we reproduced results on baseline architectures from the cited paper, [‡] indicates models in the close vicinity of the optimum. Results are averaged over 5 runs.

## Broader Impact

We introduce Collegial Ensembles (CE), which is a principled framework for neural network architecture search. The paper makes theoretical contributions backed up by empirical results. In addition, there is a practical message: In order to find a good model with a fixed budget of parameters, one would optimize the primal form of the CE objective, which will minimize variance of the NTK (maximize smoothness) and thereby have a better chance of improving generalization performance. This essentially aims to make optimal use of fixed model capacity. Instead, if the goal is to find a model that minimizes the number of parameters of a base architecture while preserving performance, one would optimize the dual form. This is a form of principled model pruning backed by theory and empirical analysis.

Crucially, the CE method allows a user to make smart choices to construct neural network models without needing to train a large search space of models. Instead, they can plug a base architecture into a simple formula corresponding to either the dual or primal objective, and get back a reasonable model that only needs to be trained once. Given that the typical architecture selection process involves an expensive grid search or evolutionary strategy, advances like CE have the potential to reduce data center costs and developer time.

In terms of scientific contributions, CE introduces theory and analysis that furthers the growing body of work on understanding deep learning in wide networks, including insights on the neural tangent kernel, which is becoming an important tool in analysis of neural network training behavior. In addition, this work leads to insights in how capacity can be smartly used to produce improved generalization on held out data, which is still a poorly understood aspect of machine learning – which has traditionally focused on optimization. Finally, the empirical studies demonstrate that the theoretical framework can explain certain design choices that have become goto neural network architectures including ResNext, which can lead to further principled advancements.

Considering that our work is fundamental science, we are not advocating a particular application, and are instead trying to ensure that the practitioner has more control over how to effectively design models.

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
