[Supplementary Material]

# Appendix: Collegial Ensembles

**Etai Littwin   Ben Myara   Sima Sabah   Joshua Susskind   Shuangfei Zhai  Oren Golan**
Apple Inc.
{elittwin, bmyara, sima, jsusskind, szhai, ogolan}@apple.com

## 1   Results and Implementation Details: Downsampled ImageNet

### 1.1   Experiments on ImageNet32×32

**Implementation details.** We use the same ResNeXt-29 architectures from the CIFAR experiments. We use SGD with $0.9$ momentum and a batch size of $1024$ on 8 GPUs ($128$ per GPU). The weight decay is $0.0005$ and the initial learning rate $0.08$. We train the models for $80$ epochs and divide the learning rate by a factor $5$ at epoch $20$, $40$ and $60$. We use the same data normalization and augmentations as in [1].

**Primal formulation.** Using the same baseline architecture $1{\times}128$d as for the CIFAR experiments, we train the model achieving the optimum, $37{\times}10$d, and report results in Table 1. Our optimal model achieves lower top-1 and top-5 errors than the baseline ResNeXt architecture $3{\times}64$d derived in [5] at a similar parameter budget. We use the same augmentations and learning rate schedule as [1].

**Dual formulation.** Using the same baseline $1{\times}128$d and optimally compact architecture $10{\times}10$d derived for the CIFAR experiments, we observe a similar trend: our optimal model suffers lighter top-1 and top-5 degradation than the Wide ResNet variant with a reduced parameter budget, with $2.8$ times fewer parameters than the baseline. Sampling models on the dual curve with lower $\rho$ such as $3{\times}40$d, we find models that suffer less than a percent drop in top-1 and top-5 error with a significantly lower parameter count.

| | $\rho$ | Params | Top-1 error | Top-5 error |
|---|---|---|---|---|
| Wide ResNet 28-10 [1] | - | 37.1M | 40.96 | 18.87 |
| ResNet-29, $1{\times}128$d[†] | 1 | 14.8M | 40.61 | 17.82 |
| ResNeXt-29, $3{\times}64$d[†] | 1 | 14.4M | 39.58 | 17.09 |
| ResNeXt-29, $37{\times}10$d (Ours) | 1 | 14.8M | **38.41** | **16.13** |
| Wide ResNet 28-5 [1] | 1.6 | 9.5M | 45.36 | 21.36 |
| ResNeXt-29, $10{\times}10$d (Ours) | **2.8** | **5.2M** | 43.36 | 19.65 |
| ResNeXt-29, $3{\times}40$d (Ours) | 1.8 | 8.0M | 41.54 | 18.58 |

Table 1: **Primal and dual results for ResNet baselines on ImageNet32×32**. Top-1 and top-5 errors (in %) and model sizes are reported. The optimally smooth model, $37{\times}10$d, surpasses the baseline architectures from [5] (indicated with [†]) with the same number of parameters. The optimally compact model, $10{\times}10$d, achieves slightly degraded results but with $2.8$ times fewer parameters. Results are averaged over 5 runs.

### 1.2   Experiments on ImageNet64×64

**Implementation details.** In order to adapt the ResNeXt-29 architectures used for CIFAR-10/100 and ImageNet32×32 to the resolution of ImageNet64×64, we add an additional stack of three residual blocks following [1]. Following the general parametrization of ResNeXt [5], we multiply the width of this additional stack of blocks by $2$ and downsample the spatial maps by the same factor using a strided convolution in the first residual block. We use SGD with $0.9$ momentum and a batch size of

512 on 8 GPUs (64 per GPU). The weight decay is 0.0005 and the initial learning rate 0.04. We train the models for 60 epochs and divide the learning rate by a factor 5 at epoch 20, 30 and 40. We use the same data normalization and augmentations as [1].

| | $\rho$ | Params | Top-1 error | Top-5 error |
|---|---|---|---|---|
| Wide ResNet 36-5 [1] | - | 37.6M | 32.34 | 12.64 |
| ResNet-38, $1\times96$d$^\dagger$ | 1 | 37.5M | 29.36 | 10.10 |
| ResNeXt-38, $2\times64$d$^\dagger$ | 1 | 39.0M | 28.86 | 9.72 |
| ResNeXt-38, $22\times10$d (Ours) | 1 | 36.3M | **28.34** | **9.38** |
| ResNeXt-38, $8\times10$d (Ours) | **2.3** | **16.3M** | 30.93 | 11.02 |
| ResNet-38, $1\times128$d$^\dagger$ | 1 | 57.8M | 28.56 | 9.67 |
| ResNeXt-38, $3\times64$d$^\dagger$ | 1 | 56.1M | 28.01 | 9.28 |
| ResNeXt-38, $37\times10$d (Ours) | 1 | 57.7M | **27.24** | **8.74** |
| ResNeXt-38, $10\times10$d (Ours) | **3.0** | **19.1M** | 30.22 | 10.60 |

Table 2: **Primal and dual results for ResNet baselines on ImageNet64×64.** Top-1 and top-5 errors (in %) and model sizes are reported. The optimally smooth models, 22×10d and 37×10d, surpass the baseline architectures from [5] (indicated with $^\dagger$) with the same number of parameters. The optimally compact models, 8×10d and 10×10d, achieve slightly degraded results but with significantly fewer parameters. Results are averaged over 5 runs.

## 2 Results on CIFAR-10

Results are shown in Fig. 1 and implementation details can be found in the main text in Sec. **??**.

(a) Primal                    (b) Dual

Figure 1: **Test errors (in %) on CIFAR-10.** Clear correlation is observed between test error and the primal curve. (a) Different models sitting on the primal curve. The model achieving the primal minimum, 37×10d, achieves the best error. (b) Different models sitting on the dual curve. The model 10×10d achieves the dual maximum ($\rho \approx 3.3$) and a slightly higher test error than the baseline with 3.3 times fewer parameters. Results are reported over 10 runs and shown with standard error bars.

## 3 FLOPs efficiency

Throughout the paper, we considered the parameter efficiency $\rho$ defined as the ratio between the number of parameters of the baseline model and the ensemble. Using this definition of efficiency, models satisfying the primal objective were models with similar number of parameters. Instead of using the parameter efficiency, we can consider FLOPs efficiency in the same way:

$$\rho^{\text{flop}}(m, \mathbf{n}) \triangleq \frac{\beta_s^{\text{flop}}}{\beta_e^{\text{flop}}(\mathbf{n})} = \frac{\beta_s^{\text{flop}}}{m\beta^{\text{flop}}(\mathbf{n})}, \tag{1}$$

where $\beta_s^{\text{flop}}$ and $\beta_e^{\text{flop}}$ are the number of FLOPs of the baseline model and the total number of FLOPs in the ensemble respectively. We report results for the primal formulation on CIFAR-10/100 in Table 3. We see that the model achieving the primal optimum, $44{\times}8$d, attains better test error on CIFAR-10 and CIFAR-100 with similar number of FLOPs.

| Model | GFLOPs | Params | C10 | C100 |
|---|---|---|---|---|
| ResNet-29 $1{\times}128$d [5][†] | 4.19 | 13.8M | 4.08 | 19.39 |
| ResNeXt-29 $3{\times}64$d [5][†] | 4.15 | 13.3M | 3.96 | 18.57 |
| ResNeXt-29 $44{\times}8$d (Ours) | 4.20 | 12.7M | **3.66** | **17.86** |
| ResNeXt-29 $60{\times}6$d (Ours)[‡] | 4.17 | 12.6M | 3.73 | 18.04 |

Table 3: **Results for ResNeXt-29 baselines on CIFAR-10/100 when keeping FLOPs constant instead of # parameters**. Test errors (in %) for CIFAR-10 (C10) and CIFAR-100 (C100) along with model GFLOPs and sizes are reported. The optimally smooth model, $44{\times}8$d, surpasses the baselines with the same number of FLOPs. [†] indicates we reproduced results on baseline architectures from the cited paper, [‡] indicates models in the close vicinity of the optimum. Results are averaged over 10 runs.

# 4 Fitting $\alpha$ to a ResNet Block

Figure 2: **Estimating the variance of a ResNet block and fitting $\alpha$**. The Monte Carlo estimate is calculated over 2000 trials and $\alpha$ is fitted following Algorithm 1.

# 5 Proofs of Lemma 1 and Theorem 1

**Lemma 1** (Infinite ensemble). *The following holds:*

$$\mathcal{K}^e(\Theta) \xrightarrow{a.s} \mathcal{K}_\mathbf{n}^\infty \quad \text{as } m \to \infty. \tag{2}$$

*Proof.* Recall that the NTK of the ensemble is given by the mean:

$$\mathcal{K}^e(X;\Theta) = \frac{1}{m}\sum_{j=1}^{m}\mathcal{K}_\mathbf{n}(X;\theta_j). \tag{3}$$

Note that expectation of each member in the average is identical and finite under Lebesgue integration:

$$\mathcal{K}_\mathbf{n}^\infty(X) = \mathbb{E}_\theta[\mathcal{K}_\mathbf{n}(X;\theta_j)]. \tag{4}$$

Since each member of the ensemble is sampled independently, we have from the strong law of large numbers (LLN):

$$\frac{1}{m}\sum_{j=1}^{m}\mathcal{K}_\mathbf{n}(X;\theta_j) \xrightarrow{a.s} \mathcal{K}_\mathbf{n}^\infty(X) \quad \text{as } m \to \infty. \tag{5}$$

Proving the claim. □

**Theorem 1** (NTK evolution over time). *Assuming analytic activation functions $\phi(\cdot)$ with bounded derivatives of all orders, and the $L_2$ cost function , it holds for any finite $t$:*

$$\mathcal{K}^e(\Theta_t) - \mathcal{K}^e(\Theta_0) \sim \mathcal{O}_p(\frac{1}{mn}) \tag{6}$$

*where the notation $x_n = \mathcal{O}_p(y_n)$ states that $x_n/y_n$ is stochastically bounded.*

*Proof.* In the following proof, we assume for the sake of clarity our training data contains a single example, so that $\mathcal{K}_n, \mathcal{K}^e, \mathcal{F}_n, \mathcal{F}^e \in \mathcal{R}$. The results however hold in the general case. Throughout the proof, we use $\Theta_t$ to denote the weights at time $t$, while $\Theta, \theta_j$ denote the weights at initialization.

For analytic activation functions, the time evolved kernel $\mathcal{K}^e(\Theta_t)$ is analytic with respect to $t$. Therefore, at any time $t$ we may approximate the kernel using Taylor expansion evaluated at $t = 0$:

$$\mathcal{K}^e(\Theta_t) - \mathcal{K}^e(\Theta) = \sum_{r=1}^{\infty} \frac{\partial^r \mathcal{K}^e(\Theta)}{\partial t^r} \frac{t^r}{r!}. \tag{7}$$

Similarly to the technique used in [4], we assume we may exchange the Taylor expansion with large width and multiplicity limits. We now analyze each term in the Taylor expansion separately. Using the ensemble NTK definition in the main text, the $r$-th order term of the ensemble NTK is given by:

$$\frac{\partial^r \mathcal{K}^e(\Theta)}{\partial t^r} = \frac{1}{m} \sum_{j=1}^{m} \frac{\partial^r \mathcal{K}_n(\theta_j)}{\partial t^r} = \frac{1}{m} \sum_{j=1}^{m} \left(\frac{\partial}{\partial t}\right)^r \mathcal{K}_n(\theta_j). \tag{8}$$

Next we derive the time derivative operator $\frac{\partial}{\partial t}$, under gradient flow and $L_2$ loss. Given label $y$, we can denote the residual terms $\mathcal{R}_j = \left(\mathcal{F}_n(\theta_j) - \frac{y}{\sqrt{m}}\right) \in \mathcal{R}^N$ and the total model residual as $\mathcal{R}(\Theta) = \frac{1}{\sqrt{m}} \sum_{j=1}^{m} \mathcal{R}_j$, such that the $L_2$ cost given by $\mathcal{L} = \frac{1}{2}\left(\mathcal{R}(\Theta)\right)^2$. The model parameters in this case evolve according to:

$$\dot{\Theta} = -\nabla_\Theta \mathcal{L} = -\left(\frac{\partial \mathcal{F}^e(\Theta)}{\partial \Theta}\right) \mathcal{R}(\Theta). \tag{9}$$

The parameters of each model $j$ in the ensemble evolve according to:

$$\dot{\theta}_j = -\nabla_{\theta_j} \mathcal{L} = -\left(\frac{\partial \mathcal{F}^e(\Theta)}{\partial \theta_j}\right) \mathcal{R}(\Theta) = -\frac{1}{\sqrt{m}}\left(\frac{\partial \mathcal{F}_n(\theta_j)}{\partial \theta_j}\right) \mathcal{R}(\Theta). \tag{10}$$

The time derivative operator $\frac{\partial}{\partial t}$ at $t = 0$ can be expanded as follows:

$$\frac{\partial}{\partial t} = \left\langle \dot{\Theta}, \frac{\partial}{\partial \Theta} \right\rangle = \sum_{j=1}^{m} \left\langle \dot{\theta}_j, \frac{\partial}{\partial \theta_j} \right\rangle \tag{11}$$

Plugging the definition of $\dot{\theta}_j$ in Eq. 10 into Eq. 11 yields:

$$\frac{\partial}{\partial t} = -\frac{1}{\sqrt{m}} \mathcal{R}(\Theta) \sum_{j=1}^{m} \left\langle \frac{\partial \mathcal{F}_n(\theta_j)}{\partial \theta_j}, \frac{\partial}{\partial \theta_j} \right\rangle = -\frac{1}{\sqrt{m}} \mathcal{R}(\Theta) \sum_{j=1}^{m} \hat{\Gamma}_j \tag{12}$$

where we have introduced the operator $\hat{\Gamma}_j = \langle \frac{\partial \mathcal{F}_n(\theta_j)}{\partial \theta_j}, \frac{\partial}{\partial \theta_j} \rangle$. For each model $j = j_0$ in the ensemble, the $r$-th time derivative of its corresponding NTK at $t = 0$ is therefore given by:

$$\left(\frac{\partial}{\partial t}\right)^r \mathcal{K}_n(\theta_{j_0}) = \left[-\frac{1}{\sqrt{m}}\mathcal{R}(\Theta)\left(\sum_{j=1}^{m} \hat{\Gamma}_j\right)\right]^r \mathcal{K}_n(\theta_{j_0}) \tag{13}$$

$$= \left[-\frac{1}{m}\left(\sum_{j_1,j=1}^{m} \mathcal{R}_{j_1}\hat{\Gamma}_j\right)\right]^r \mathcal{K}_n(\theta_{j_0}) \tag{14}$$

Denoting $\overset{\frown}{\prod}_{u=1}^{r} A_u = A_r A_{r-1}...A_1 = \left[A\right]^r$, we can now expand the $r$-th derivative:

$$\left(\frac{\partial}{\partial t}\right)^r \mathcal{K}_n(\theta_{i_0}) = \left(\frac{-1}{m}\right)^r \sum_{j_1...j_r=1}^{m} \left[\overset{\frown}{\prod}_{u=1}^{r}\left(\mathcal{R}_{j_u}\sum_{j=1}^{m}\hat{\Gamma}_j\right)\right]\mathcal{K}_n(\theta_{j_0}) = \left(\frac{-1}{m}\right)^r \sum_{j_1...j_r=1}^{m} \xi_{j_0...j_r} \tag{15}$$

where $\xi_{j_0\ldots j_r} = \left[\overset{\curvearrowright r}{\prod_{u=1}} \left(\mathcal{R}_{j_u} \sum_{j=1}^m \hat{\Gamma}_j\right)\right] \mathcal{K}_n(\theta_{j_0})$.

Using the notation $\forall_{j,u}, \quad (\hat{\Gamma}_j)^u \mathcal{R}_j = \mathcal{R}_j^{(u)}$, and noticing that $\mathcal{K}_n(\theta_j), \mathcal{R}_j$ depend only on $\theta_j$, the following hold:

$$\forall_{j_1 \neq j_2}, \quad \hat{\Gamma}_{j_1} \mathcal{K}_n(\theta_{j_2}) = 0. \tag{16}$$

$$\forall_{u,j_1 \neq j_2}, \quad \hat{\Gamma}_{j_1} \mathcal{R}_{j_2}^{(u)} = 0. \tag{17}$$

$$\forall_j, \quad \hat{\Gamma}_j \mathcal{R}_j = \mathcal{K}_n(\theta_j). \tag{18}$$

$$\forall_{j,u,v}, \quad \hat{\Gamma}_j \mathcal{R}_j^{(u)} \mathcal{R}_j^{(v)} = \mathcal{R}_j^{(u+1)} \mathcal{R}_j^{(v)} + \mathcal{R}_j^{(u)} \mathcal{R}_j^{(v+1)}. \tag{19}$$

where Eq. 19 is the application of the chain rule.

Using the above equalities, the terms $\xi_{j_0\ldots j_r}$ can be expressed as a sum over a finite set $\mathbf{S}^r$ as follows:

$$\forall_{j_0,,,j_r}, \quad \xi_{j_0\ldots j_r} = \sum_{\mathbf{S}^r} \prod_{v=0}^r \mathcal{R}_{j_v}^{(u_v)}, \quad \mathbf{S}^r := \{u_v\}_{v=0}^r \Big|_{\substack{\forall_{0<v}, 0 \leq u_v \leq r-v \\ 2 \leq u_0 \leq r+1 \\ \sum_{v=0}^r u_v = r+1}} \tag{20}$$

**Example:** for $r = 2$, the term $\xi_{j_0,j_1,j_2}$ is expanded as follows:

$$\xi_{j_0,j_1,j_2} = \left[\overset{\curvearrowright 2}{\prod_{u=1}} \left(\mathcal{R}_{j_u} \sum_{j=1}^m \hat{\Gamma}_j\right)\right] \mathcal{K}_n(\theta_{j_0}) \tag{21}$$

Expanding the multiplication and using Eq. 18:

$$\xi_{j_0,j_1,j_2} = \left(\mathcal{R}_{j_2} \sum_{j=1}^m \hat{\Gamma}_j\right)\left(\mathcal{R}_{j_1} \sum_{j=1}^m \hat{\Gamma}_j\right) \mathcal{R}_{j_0}^{(1)} \tag{22}$$

Using the chain rule in Eq.19, and eliminating elements using Eq. 17:

$$\xi_{j_0,j_1,j_2} = \left(\mathcal{R}_{j_2} \sum_{j=1}^m \hat{\Gamma}_j\right) \mathcal{R}_{j_1} \mathcal{R}_{j_0}^{(2)} = \mathcal{R}_{j_2} \mathcal{R}_{j_1}^{(1)} \mathcal{R}_{j_0}^{(2)} + \mathcal{R}_{j_2} \mathcal{R}_{j_1} \mathcal{R}_{j_0}^{(3)} \tag{23}$$

We can now express the result in the formulation of Eq. 20

$$\xi_{j_0,j_1,j_2} = \sum_{\mathbf{S}^2} \prod_{v=0}^2 \mathcal{R}_{j_v}^{(u_v)}, \quad \mathbf{S}^2 := \{u_v\}_{v=0}^2 \Big|_{\substack{\forall_{0<v}, 0 \leq u_v \leq 2-v \\ 2 \leq u_0 \leq 3 \\ \sum_{v=0}^2 u_v = 3}} \tag{24}$$

The $r$'th time derivative of the full ensemble $\mathcal{K}^e$ is given by:

$$\left(\frac{\partial}{\partial t}\right)^r \mathcal{K}^e = \frac{1}{m} \sum_{j_0=1}^m \left(\frac{\partial}{\partial t}\right)^r \mathcal{K}_n(\theta_{j_0}) \tag{25}$$

$$= \frac{(-1)^r}{m^{r+1}} \sum_{j_0\ldots j_r=1}^m \xi_{j_0\ldots j_r} \tag{26}$$

$$= \frac{(-1)^r}{m^{r+1}} \sum_{j_0\ldots j_r=1}^m \sum_{\mathbf{S}^r} \prod_{v=0}^r \mathcal{R}_{j_v}^{(u_v)} \tag{27}$$

$$= (-1)^r \sum_{\mathbf{S}^r} \prod_{v=0}^r \left(\frac{\sum_{j=1}^m \mathcal{R}_j^{(u_v)}}{m}\right) \tag{28}$$

Note that for $0 \leq u$, the term $\mathcal{R}_j^{(u+1)}$ represents the $u$'th time derivative of $\mathcal{K}_n(\theta_j)$ under gradient flow with the loss $\mathcal{L} = \mathcal{R}_j$. Using results[1] from [2] on wide single fully connected models, we have

that $\forall_{u>1}\ \mathcal{R}_j^{(u)} \sim \mathcal{O}_p(n^{-1})$, and $\mathcal{R}_j^{(1)} \sim \mathcal{O}_p(1)$. Moreover, from the independence of the weights $\forall_{j_1 \neq j_2},\ \theta_{j_1} \perp\!\!\!\perp \theta_{j_2}$, it holds that $\forall_{u,j_1 \neq j_2},\ \mathcal{R}_{j_1}^u \perp\!\!\!\perp \mathcal{R}_{j_2}^u$. Therefore, for any fixed $r$ we may apply the central limit theorem for large $m$ on the terms in Eq. 28 individually:

$$\left( \frac{\sum_{j=1}^m \mathcal{R}_j^{(u)}}{m} \right) \sim \begin{cases} \mathcal{O}_p(\frac{1}{\sqrt{mn}}) & u > 1 \\ \mathcal{O}_p(1) & u = 1 \\ \mathcal{O}_p(\frac{1}{\sqrt{m}}) & u = 0 \end{cases} \tag{29}$$

Plugging back into Eq. 28 yields the desired result by noticing that $2 \leq u_0$ and $u_r = 0$:

$$\left( \frac{\partial}{\partial t} \right)^r \mathcal{K}^e \sim \mathcal{O}_p(\frac{1}{nm}) \tag{30}$$

$\square$

## 6 Figure Illustrating Theorem 1

Figure 3: Dynamics of the NTK during training as a function of width $n$ and multiplicity $m$ for (a) baseline single model (b) ensemble (c) ensemble with a constant width $n = 100$, and multiplicity $m$ to match the number of parameters in the baseline (red). The NTK was computed for a single off-diagonal entry for a depth $L = 4$ fully connected network trained on MNIST. The y axis corresponds to the absolute change in log scale between the NTK value at initialization, and after training for $T = 100$ epochs. As predicted in Theorem 1, the baseline model with $m = 1, n = d^2$ and the ensemble with $m = n = d$ have equal $mn$, therefore exhibit similar correction of the NTK. In (c), the change of the NTK becomes smaller than the baseline, as mn is considerably larger, although the total parameter count is the same as the baseline.

## Footnotes

[1] In [2], the $\mathcal{O}_p(n^{-1})$ result was obtained using a conjecture, and demonstrated empirically to be tight. An $\mathcal{O}_p(n^{-0.5})$ result of the same quantity is obtained rigorously in [3], which yields an asymptotic behaviour of $\mathcal{O}_p(n^{-0.5} m^{-1})$ for the ensemble.