[Reviews · NeurIPS 2020]

Review 1

Summary and Contributions: The paper considers ensemble models, treating the number of models in an ensemble as an analog to the width of a network. It shows that (1) such ensemble models can lead to improved performance compared with a single (wider) model that has the same number of parameters, and that (2) such ensemble models can be constructed with far fewer parameters without significant loss in performance. This is achieved by relying on large width theory. It is assumed that having lower kernel variance is beneficial, namely that kernel variance good proxy for test error. With this assumption, one can (1) find the combination of individual model width and ensemble size that minimizes the variance at a given number of parameters, and (2) find the combination that minimizes number of parameters at a given variance. Thus, one can construct models that are more performant or are more compact.

Strengths: One of the main challenges of applying large width theory to practice is the fact that the number of parameters grows quadratically with width. This is a problem especially for convolutional networks, where the memory requirements quickly become infeasible as one increases the number of channels. This work suggests a practical method of harnessing the power of large width networks while only paying a linear cost in number of parameters, because this number grows linearly with the number of models in the ensemble. While the method is in principle independent of large width theory, the authors find that large width theory is beneficial in finding the optimal combination of model width and number of ensemble models. This relies on the assumption that kernel variance is a good proxy for test performance. This assumption is borne out in Figures 4 and 6.

Weaknesses: The assumption that the kernel variance is a good proxy for the test error is strong. In particular, empirical evidence suggests that modern networks are trained far from the NTK regime, so it is not clear what role the kernel plays in the training dynamics of such networks. However, Figure 6 (a) does show that this assumption holds! It would be great if the authors could either shed light on why they believe this assumption holds, or (if they cannot) point out that it is a surprising result. To be clear, it may be beyond the scope of this paper to answer this question, but I think it is at least worth commenting on.

Correctness: I believe the claims made in the paper; the theoretical predictions are backed up by a large number of experiments.

Clarity: Yes.

Relation to Prior Work: Yes.

Reproducibility: Yes

Additional Feedback: Nits: Equation 1 and the fact that the kernel is constant only holds in the gradient flow limit. With gradient descent, significant deviation can occur depending on the learning rate. Line 110: I assume $K_\infty$ is the infinite width limit of the kernel, but as far as I can tell it is not explicitly defined.


Review 2

Summary and Contributions: The paper introduces the concept of collegial ensembles motivated by recent developments in theoretical study of overparameterized networks. Collegial ensemble is an aggregation of multiple independent models with identical architecture. With the concept of collegial ensemble and their connection to Neural Tangent Kernel (NTK), authors propose a practical method to find either a better performing model with same parameter count or similarly performing model with much less number of parameters than baseline model. They show the protocol works for ResNext where grouped convolution can be naturally connected to a collegial ensemble.

Strengths: The concept of collegial ensemble may not be so novel but the author’s finding making connection to the theory of NTK and grouped convolution is quite interesting. Paper’s theoretical analysis on the collegial ensemble is sound and clear (Section 2). While the proposed search procedure for an efficient ensemble is not fully grounded with theory, the paper’s experiments on ResNext are impressive. The paper would be interesting to both theoretical deep learning communities and researchers who are interested in practical architecture design or search. It would be likely leading to various follow up work both studying the why collegial ensemble works well and also applying it to find better or efficient neural network architecture. Also the authors included well documented and well structured code which is encouraging for reproducibility perspective.

Weaknesses: Protocol for finding the efficient ensemble in Section 3 is based on a hypothesis with limited checks. Proposition 1, as far as I know, is proved for the diagonal element of NTK for fully connected ReLU networks when both width and depth grows simultaneously. The practical setting the paper is considering is quite far from this limit and the paper would have been stronger if the test on the hypothesis was done more thoroughly beyond just Figure 8. While ablation study does highlight the effect of variance and capacity, readers would be more convinced if similar behavior was shown for other workloads (at least on more complicated dataset). For the reviewer, claim in Fig 4 that increasing capacity without reducing variance leading to bad performance is hard to read from the plot when the errors are already quite small.

Correctness: Theoretical analysis connecting to NTK is correct. The empirical methodology is sound although based on a heuristic.

Clarity: The paper is clearly structured and presented and for most part the results are easy to understand from the figure and the tables. Few minor nits included in the additional feedback.

Relation to Prior Work: The paper does a good job of describing previous work and describes current work’s contribution. Only addition I would suggest is to discuss similar findings in section 4 of [1] and Figure 2 of [2] regarding ensemble and width effects contributing similarly to kernel convergence for both NTK and GP kernels. [1] Novak et al., Bayesian Deep Convolutional Networks with Many Channels are Gaussian Processes, ICLR 2019 [2] Novak et al., Neural Tangents: Fast and Easy Infinite Neural Networks in Python, ICLR 2020

Reproducibility: Yes

Additional Feedback: [Post Author Response] I have read the author response and other reviews. I think the paper is a strong paper and generally happy with authors response. Still as R3 points out theoretical justification of why var(K) is quantity to minimize is to be desired. Given the practicality demonstrated on ResNext on ImageNet, I think the CE could generalize to practical settings and still will keep my original score. -------------------------------------------- I would encourage the authors to expand on empirical verification of Proposition 1 for practical architectures. Question: Is there a reason that main ImageNet results are not highlighted in the main text? The gain in the primal perspective is not great but since ImageNet architecture is overly fintuned even the dual perspective parameter efficiency seems to be a good result. I wonder why authors decide to focus more on results based on CIFAR-10/100. Few nits L38,59, 81,118, 227,.. : fix the quote direction l49 : neutral -> neural l68: two periods


Review 3

Summary and Contributions: The authors study ensemble of over-parameterized models and propose a method to find better performant model under a fixed budget (total number of parameters ). More precisely, the problem the authors want to address is how to balance m (the number of ensemble) and n (roughly, the width of the network) so that (1) the total number of params is roughly the same, and (2) the model achieves best performance. The authors propose minimizing the variance of the neural tangent kernel (NTK) of the ensemble network and support their proposal via experiments.

Strengths: The idea seems pretty interesting. Experiment section also provides evidence to support their proposal. The method could be useful for efficient/more principled architectures search if its effectiveness is verified via large scale experiments and more challenging tasks.

Weaknesses: The main issue I see is it is unconvincing why minimizing var(K) is the right measurement for better generalization. No theoretical support for this point in the paper. Minimizing var(K) seems to push the network towards the kernel regime (e.g. var(K) ==0 in the exact kernel regimes). This also seems to be at odd with many recent works that argue *lazy training* (i.e. the kernel regime) is NOT doing representation learning and hurts the performance of the network. Practical models are definitely operating outside the kernel regime. In addition, [1] also argues that var(K) does not vanish might be helpful for ``features learning``. Of course, if var(K) is large, the network will be difficult to train. Therefore, I am more convinced that var(K) is a good measure for trainability but not convinced it is a good measurement for generalization at the moment. Also, the correctness of Proposition 1, on which the main argument of the paper rely, hasn't been proved yet. To the best of my knowledge, proposition 1 is only proved for the diagonal terms for Relu network [1], not for off-diagonals or other activation functions. [1] Boris Hanin and Mihai Nica. Finite depth and width corrections to the neural tangent kernel.

Correctness: The overall results seem correct to me.

Clarity: Yes. The paper is well written and easy to follow.

Relation to Prior Work: Yes.

Reproducibility: Yes

Additional Feedback: I increase my score to 6 weakly accept because I think the practical value of the paper outweighs its weak point in theory. Other comments. 1. CE is essentially a variance reduction trick. It will be helpful see how the spectrum of the NTK changes with fixed number of params, but changing m and n. In particular, investigating the variance of the condition number of the NTK (or largest/smallest singular values). The library https://github.com/google/neural-tangents provides tool to compute the finite width NTK. 2. Since we agree that var(K) is a proxy for trainability, it will be good to see how an optimal CE improves training speed. Note that, CE could potentially reduce the largest singular value of the NTK which allows a larger learning rate comparing to the baseline model. 3. Both https://arxiv.org/abs/1909.08156 and https://arxiv.org/abs/1909.11304 obtained O(1/n) for the change of the NTK during training, which could translates to the 1/mn decay in Thm 1. Question: *after* training, how does empirical NTK(i) look like? Are they very different/similar to each other? Here, i = 1, 2, ..., m, representing different instantiation of the network. It will be interesting to look at how each NTK(i) evolves during training. ------------------------------- 1. Since Proposition 1 hasn't been proved yet, it shouldn't called a *proposition* (or lemma, etc). Calling it an *Assumption* might be more proper here. 2. Lemma 1 uses the strong law of large number. Technically, one needs to verify the finiteness of the first moment, i.e. eq (5) is finite, which should be quite straightforward. 3. Theorem 1 seems not quite related to the main theme of the paper and it is not relevant to the main experimental section at all. Or do I miss something? The NTK of performant model usually moves quite far away from its initialization.


Review 4

Summary and Contributions: This paper introduces Collegial Ensemble (CE), which is the average of $m$ networks with identical architectures, each with width $n$. Authors first presented a theoretical analysis of CE in terms of its kernel function and its evolution (i.e., the bound on the parameter drift from initialization) in the asymptotic limit. Then, author presented a method for searching for optimal $m$ and $n$ by either minimizing the asymptotic bound of kernel variance while holding parameter efficiency constant (the primal formulation), or by maximizing parameter efficiency while holding model variance constant (the dual formulation). Author validated their approach on toy datasets (MNIST) and on ResNeXt for CIFAR and reduced ImageNet.

Strengths: This work provides two contributions: (1) asymptotic analysis of CE, (2) an algorithm for ensemble architecture search based on a constrained optimization problem for parameter efficiency / model variance. While the theoretical analysis (contribution 1) is standard, contribution 2 is in fact both significant and novel. It presents an principled (and seemingly practical) approach for specifying model architecture for ensemble models that draws connection with the recent result of model variance in the overparametrization regime. I believe this work nicely leverages recent advances in kernel analysis to bring a simple solution to a relevant problem (i.e., architecture search). Therefore I believe it will be relevant for the NeurIPS community.

Weaknesses: Major Comments: (1) Organization of theoretical results. * It is not clear to me the significance of Lemma 1 and Theorem 1, specifically, why they are surprising or important for CE (e.g., lemma 1 seem to be a standard application of LLN), and what insight can be derived from them (e.g., why should I care about the asymptotic rate of the stochastically bound O(1/nm)? Does it related to any operating characteristics of the model?). Comparing to these results, Proposition 1 seem to be more central to this paper, and may merit more detailed explanation. If authors agree, it may be beneficial to shift the emphasis in the presentation of the theoretical results, for example, move Lemma 1 to the Appendix. Unless Theorem 1 has crucial implications that is central to the story of the paper, I suggest author move it to the appendix also, or alternatively give a bit more explanation to justify its position in the paper. * As mentioned in the last paragraph, Proposition 1 seem to be more important and deserves its own background section. In particular, author should explain what $\alpha$ in Equation (2) represents, and explain how to compute it in theory and in practice, thereby providing context for Algorithm 1. It will also reader to understand the importance of Proposition 1 by explaining the connection between Var(K) and model's generalization ability, like what you illustrated empirically in Figure 4. (2) (Optionally,) adding baseline comparison to experiments 4.2, and more discussion. While I find the experiments interesting, it might be beneficial to add at least one standard baseline method (e.g., standard neural architecture search) to compare against primal formulation. So author can compare the difference in terms of resulting architecture, estimated kernel variance, testing error and total wall clock time used to better illustrate the benefit of the existing method. Minor: * line 68, "respectively.." (should be only one period) * line 148 cifar10/100 -> CIFAR-10/-100 * There's some relevant recent work on NTK for ensemble models, e.g., [1] and references therein. It appears rather relevant and should be included in Section 5 as well. [1] Bobby He, Balaji Lakshminarayanan, Yee Whye Teh. Bayesian Deep Ensembles via the Neural Tangent Kernel. (https://arxiv.org/abs/2007.05864)

Correctness: Yes. Both proof and empirical methods appears to be sensible.

Clarity: The paper is reasonably clearly written, however it appears that the materials can be better organized. See Major Comments.

Relation to Prior Work: Yes. Although there are some missing literature, see Minor Comments.

Reproducibility: Yes

Additional Feedback: (Posted after author rebuttal) Thank authors for the feedback. I maintain a score of 6 on below grounds: (Positive) The paper presents a interesting theoretical study of ensemble learning from NTK perspective. I believe Theorem 1 is of sufficient significance after reading rebuttal and reference [B]. This together with the contribution 2 (see Strengths) forms a worthwhile contribution to the community. (Negative, mostly above clarity) 1. I believe it is important to provide some justification/discussion in the main text regarding the connection between var(K) and generalization. While the exact correspondence can be difficult to establish, authors can at least discuss that the var(K) contributes to the loss term (via a bias-variance decomposition of bregman loss). Otherwise the paper can feel incomplete given the central position of var(K) play in the methods. 2. To ensure the accessibility of the paper, I urge the authors to add details for Proposition 1 and Theorem 1 as discussed in the rebuttal.

[Author Response · NeurIPS 2020]



Figure I: (A) MLP $\alpha$ fitting. (B + C) Training loss curves at early and late stages for optimal primal/dual ResNext models (orange and green) and baseline (blue), trained on CIFAR100 (for details see section 4.2.1 in the main text)

**R1+R2+R3+R4:** We would like to thank the reviewers for their thoughtful comments, suggestions and constructive feedback. Typos and missing citations as pointed out by all reviewers will be addressed in future versions. The main point of contention articulated by all 4 reviewers is the validity of var(K) as a good proxy for generalization and the correctness of Proposition 1 in the general case, and so we will address these in detail. **var(K) as a proxy for generalization:** We agree with the reviewers that this is a strong assumption, and our main hypothesis is indeed to use var(K) as a proxy for *trainability*, as stated in the abstract (line 12) and introduction. As a direct evidence, we additionally show in Fig. I above that a lower training loss is achieved with the primal formulation, and the loss is roughly maintained in the dual formulation. In the ablation study and section 2, we empirically show how lower v(k) translates to better performance in the tested models under practical scenarios. We will eliminate the claims regarding v(k) and generalization in the main text, and instead leave the precise characterization of it as future work. Meanwhile, we mention [A] where a rigorous connection is made between v(k) and generalization (see section 3), advocating for ensembling as variance reducing alternatives. **Validity of Proposition 1:** Additional experiments on its validity (including Fig. I(A) above) will strengthen the paper, and will be added to the appendix. Note that in the context of section 2, its correctness in general is less important than its usefulness. In fact, any formula v(k) = f(n) that can be nicely fitted to Monte Carlo simulations in algorithm 1 can be used to derive optimal ensembles. We find that Proposition 1 holds nicely in general. **R1:Undefined notations.** $K_\infty$ is indeed the infinite width kernel. We will make this explicit in future versions. **R1:Gradient flow assumption.** We agree with the reviewer. However with some additional work the $\mathcal{O}((nm)^{-1})$ results can be derived in the infinite width limit for gradient descent with a small learning rate. We leave this extension to future work. **R2:Ablation study.** The ablation study in the paper is conducted on mnist, however figures 6,7 clearly demonstrate the effect of variance and capacity on larger datasets with modern architectures. Note that we do not make the claim that capacity in itself is harmful in the general case. **R2:Full Imagenet results.** We were unable to produce the full Imagenet results in time for the deadline of the main text, and so they were included in the appendix. In future versions they will be included in the main text. **R3: Connection of var(k) to generalization.** We do not see our results being at odds with previous work since we are not advocating for eliminating variance all together or using small learning rates. (thus eliminating the prospects of representation learning). Please see our above response for further details. **R3:Proposition 1 as an assumption.** We agree with the reviewer and will change its definition to an assumption in future versions. **R3:Lemma 1.** The proof of Lem 1 in the appendix mentions the finiteness of first moments (line 47), and we will add more details on that matter in future versions. **R3+R4:On Lemma 1 and Theorem 1.** The discussion of NTKs in the context of collegial ensemble requires us to expand the notion of the "kernel" regime, since width can be interchanged with multiplicity (number of models in the ensemble), to which section 1 is dedicated. The O(1/m) result in The 1 is non-trivial, implying that large collegial ensembles can approach the kernel regime with linear, instead of quadratic dependence on the number of weights even after training, where the independence assumption of the ensemble models is broken. The novelty here is both technical and in the final result, and we feel will be appreciated by the community. **R4:On $\alpha$ in proposition 1.** For a given architecture, $\alpha$ represents the rate in which v(k) changes with width. Computing its theoretical value for fully connected networks is a technical challenge that is beyond the scope of the paper (see [B]). Empirically, any architecture parameterized by its width can be fitted with $\alpha$ as described in Algorithm 1. **R1+R2+R3+R4:Additional experiments that reviewers may have missed** Our proposed method can be adapted to efficiency metrics other than parameter count. In section E of the appendix we adopt a FLOPS metric instead and show how for a given FLOPS budget, optimally performant ensembles can be derived.

REFERENCES

[A] M. geiger et al: Scaling description of generalization with number of parameters in deep learning. (Journal of Statistical Mechanics Theory and Experiment 2020). [B] B. Hanin et al: Finite Depth and Width Corrections to the Neural Tangent Kernel. (ICLR 2019).


[Meta-Review · NeurIPS 2020]

This paper explores ensembles from the perspective of neural networks in the width limit. The reviewers all found that the paper is well written, technically sound and the contributions are novel and significant. There was significant discussion following the author rebuttal and multiple reviewers were willing to champion the paper for acceptance. One concern shared by reviewers was the motivation for why "var(K)" was a quantity that that should be minimized, as was presented in the theory of the paper. Overall, this seems like an exciting paper that will be of interest at the conference.